# The Dynamics of Fund Absorption: Evaluating the Efficacy of EU Structural Funds in Mitigating Regional Inequalities—Calabrian Case

**Guzmán A. Muñoz-Fernández [1,*], Angela Bertucci [1], José E. Ramos-Ruiz [2] and Maria Luisa Garo [3]**

[1] Department of Business Organization, Faculty of Law, Economics and Business Studies, University of Cordoba, 14071 Cordoba, Spain; angebertucci@gmail.com
[2] Department of Applied Economics, Faculty of Law, Economics and Business Studies, University of Cordoba, 14071 Cordoba, Spain; d22raruj@uco.es
[3] Independent Researcher, 89900 Vibo Valentia, Italy; marilu.garo@gmail.com
[*] Correspondence: guzman.munoz@uco.es

**Abstract:** The European Union aims for territorial cohesion, with human capital as a key factor. Assessing how investment in regional human capital enhances this cohesion is therefore essential. This study assesses the impact of the EU Structural Funds (ESFs) in Calabria (Italy), a region grappling with economic challenges and a brain drain phenomenon. Aimed at fostering regional cohesion, ESFs have been directed towards supporting Calabrian graduates' pursuit of master's degrees, intending to incentivize their retention or return postgraduation. A comprehensive survey of the beneficiaries of these subsidies was carried out to determine their employability in the region and the probability of the return of migrants, analyzed by logistic regression of the data. Results demonstrate a dual effect: while the quality of education and EU funding positively influence graduates to work in Calabria, acquiring advanced skills paradoxically diminishes this propensity. Moreover, although the likelihood of returning to Calabria for those working elsewhere does increase, ESF support counterintuitively reduces this probability. The findings reveal a vicious cycle; they equip graduates with high-level skills that facilitate their access to the labor market but simultaneously encourage their migration due to more favorable conditions elsewhere. It is suggested that synergies between ESF-funded policies and those supported by the European Regional Development Fund (ERDF) should be encouraged.

**Keywords:** European structural fund; human capital; cohesion policy; Calabria

## 1. Introduction

Since its establishment, the European Union (EU) has strived to fortify economic, social, and territorial cohesion, dedicating a significant portion of its budget to alleviate disparities, particularly in remote regions plagued by various challenges (European Union 2008; Gross and Debus 2018; Hawes 2014). The Cohesion Policy, the second-largest budget allocation, seeks to diminish territorial and social imbalances with differentiated support based on regional development and Gross Domestic Product (GDP) per capita (Ahner 2009; Boldrin et al. 2001; Fiaschi et al. 2018; Giua 2017). However, using GDP per capita as the sole criterion for classifying regions and allocating funds is inadequate. European territories are diverse, possessing unique territorial assets—collectively termed "territorial capital"—rooted in their respective economic, cultural, social, and environmental contexts. This capital influences Cohesion Policy fund utilization and regional investment returns (Camagni 2017; Crescenzi et al. 2020b; Fratesi and Perucca 2019; Fratesi and Percoco 2014).

Therefore, assessing Cohesion Policy effectiveness is crucial for understanding its regional development impact and how it leverages territorial capital to enhance well-being (Calegari 2021; European Parliament's Committee 2019). Despite extensive evaluations, the literature remains divided on whether the policy effectively fosters growth, convergence, and harmonious EU development (Dall'Erba and Le Gallo 2008; Notermans 2016). Conflicting findings arise from empirical, conceptual, and methodological discrepancies (Begg 2010; Dall'Erba and Fang 2017; Fratesi 2016), coupled with the policy's diverse objectives and complex regional, national, and local regulations (Bachtrögler et al. 2020; Crescenzi et al. 2020b; Medeiros and Rocha 2014).

Furthermore, economic cohesion has waned, with polarization between Europe's northwestern core and its southern and eastern peripheries, exacerbating social and territorial divergence (Notermans 2016; Farole et al. 2011). The Cohesion Policy's relationship with territorial capital is intricate: Regions with abundant capital tend to support each other, but the policy's multifaceted nature means investments do not always directly promote growth (Camagni 2017; Fratesi and Perucca 2019). Enhancing absorption capacity is key: Not just funding volume but sustainable utilization of EU funds matters (Aivazidou et al. 2020). Recent studies posit human capital and institutional quality as pivotal in achieving policy targets, linking education quality, labor market institutions, and youth unemployment reduction (Becker et al. 2013; Crescenzi et al. 2020a; Lahtinen et al. 2020; Tosun et al. 2016).

Addressing gaps in local-level effectiveness studies, our research examines the Cohesion Policy's impact on higher education and youth employment in Calabria, Italy—an Objective 1 area (Caldas et al. 2018; Giua 2017; Fratesi and Perucca 2019). The choice of Calabria for this study has two reasons. Firstly, although Calabria has natural and territorial resources like Sardinia or Provence-Alpes-Côte d'Azur and has received European funding to overcome its historical problems, it is characterized by significant structural problems that prevent foreign investment and reduce its attractiveness. Secondly, Calabria, unlike other southern Italian regions such as Puglia or Sicily, which received the same ESFs to reduce emigration flow and achieve their goals to retrain young human resources, still has a high emigration rate. We thus focused on human capital investments from 2014–2020 (Regione Calabria 2015), as evidence suggests such spending promotes economic growth (Cuaresma et al. 2018; Di Caro and Fratesi 2022).

The main objective of this research is to analyze the impacts that postgraduate scholarships, funded through the Structural Funds, have on access to the labor market and migration dynamics in the Calabria region. This analysis aims to scientifically substantiate proposals aimed at enhancing the efficacy of cohesion policy in the employment sector.

A distinctive aspect of this research, unlike previous studies, is that it does not focus on macroeconomic values. Instead, it specifically concentrates on assessing the effectiveness of these educational aids in combating unemployment and migration in the region. The purpose is to discern the local effects of investing in human capital as a tool for regional cohesion, thereby offering a more focused and in-depth perspective on its influence in the local community.

In this study, we assume that although they may have a positive impact on the integration of Calabrian graduates into the local job market, the investments of the ESFs that are aimed at improving the competencies of graduates have a modest impact at the regional level and are not sufficient to prevent the migration of the most highly qualified individuals. The methodology used to measure the outcomes of these programs was to survey graduates from the University of Calabria who received ESFs for their postgraduate studies, assessing their employment status, educational attainment, and their prospects for working in Calabria postgraduation.

This research was organized in a structured manner to achieve the main objective. Beginning with this introduction, an exhaustive bibliographic review was carried out, focusing on the Cohesion Policy and Structural Funds of the European Union as well as on the study of education and regional development. Additionally, an analysis of youth

employment in Calabria during the period 2014–2020 was conducted. Following this, the research questions guiding this study are presented, accompanied by a detailed description of the methodology used. Subsequently, an analysis of the collected data is detailed, which is subject to critical discussion. Finally, the work concludes with the presentation of the conclusions derived from the research.

## 2. Literature Review

### 2.1. Development Aid and Structural Funds in the European Union

The European Union's Cohesion Policy, executed through Structural Funds, is designed to mitigate regional disparities and bolster competitiveness and employment across Europe (European Commission 2021). Rooted in the Treaty on the European Union's objective of promoting harmonious development, these funds predominantly assist regions with less than 75% of the EU average GDP per capita. They target areas with inadequate investment, elevated unemployment, and deficient services and infrastructure (Pinho et al. 2015).

Structural Funds encompass several key components. The European Regional Development Fund (ERDF) addresses regional imbalances by fostering infrastructure development, job creation, local projects, and small business aid. The European Social Fund Plus (ESF+) has continued from the 2014–2020 cycle to uphold the European Pillar of Social Rights, preserving employment and fostering social inclusivity. The Cohesion Fund (CF) funds environmental initiatives and trans-European transport in states with a gross national income below 90% of the EU-27 average. Additionally, the Just Transition Fund (JTF), a novel instrument under the European Green Deal, facilitates regions and populations in adapting to the social and environmental repercussions of transitioning towards the EU's 2030 energy and climate targets, aspiring for a climate-neutral economy by 2050 based on the Paris Agreement (European Commission 2021).

These funds, although distinct in objectives and sectors, form an integrated financial system that bolsters economic activities in their respective regions. They enhance both tangible and intangible infrastructures, thus promoting human capital development and investment (Biedka et al. 2022; Startiene et al. 2015). The evolving perception of their efficacy in meeting Cohesion Policy goals reflects society's dynamic nature. The Structural Funds are public investments that stimulate growth and convergence in capital-scarce regions (Vukašina et al. 2022). However, exogenous factors like technological change indicate that internal shifts and economic geography (Scotti et al. 2022) cultivate a conducive environment for capital accumulation in beneficiary regions. This improves their absorptive capacity and the utilization of Structural Funds (Ciani and de Blasio 2015; Crescenzi and Giua 2016; Kersan-Škabić and Tijanić 2017).

From a neoclassical perspective, Structural Funds are vital for budgetary allocations and distribution within disadvantaged EU areas. Similarly, endogenous growth theories emphasize the long-term role of public policies in growth through human capital, innovation, and knowledge investments. When merged with economic geography theories, it becomes evident that Cohesion Policy impacts vary across European regions. This variation is contingent upon their approach to fund allocation (Marzinotto 2012; Vukašina et al. 2022). Absorption capacity stems from socio-economic and institutional development levels and each beneficiary's capability to efficiently utilize the funds (Presbitero 2016). Various impediments, whether financial, institutional, or legal, can compromise the effectiveness of the Structural Funds (Lewandowski 2023).

The scientific literature indicates a positive correlation between Structural Funds and regional development, with direct ties between EU funds and foreign investment (Caldas et al. 2018; Startiene et al. 2015) and long-term competitiveness (Tijanić and Obadic 2015). Nonetheless, factors such as high fiscal decentralization (Tosun 2014), adverse regional traits like corruption or poor governance, and inefficient administrative capacity can hinder these funds' impact. In some cases, Structural Funds may even exacerbate structural

inequalities (Czudec et al. 2019) or inadvertently disrupt income convergence within the EU (Breidenbach et al. 2019; Jagodka and Snarska 2023).

*2.2. Education, Formation, and Territorial Growth*

In recent programming cycles (2014–2020 and 2021–2027), the European Union's Structural Funds have prioritized innovation and smart growth, underscoring the critical role of education, formation, and innovation in fostering growth (Sánchez Trujillo et al. 2020). Human capital is increasingly recognized as a dynamic, lifelong process essential for cultivating basic cognitive skills and enabling individuals to acquire specialized skills (Schweisfurth and Raasch 2018), thereby mitigating the risk of skill atrophy (Dräbing and Nelson 2017).

Aligned with the Europe 2020 Strategy for growth and jobs, education emerges as a cornerstone for smart, sustainable, and inclusive growth. It facilitates access to higher education in less developed areas, enhancing employability and competitiveness and ultimately reducing poverty. The strategy sets ambitious targets, including reducing early school leavers below 10% and ensuring at least 40% of the 30–34-year-old age group attains tertiary education. It advocates for increased investment in tertiary education and lifelong learning, linking education with research and development to foster prolonged economic growth (European Commission 2010).

As of 2022, the EUROSTAT data reveal a varied distribution of education levels across Europe. Eastern countries exhibit lower proportions of individuals with low education, while countries like Spain, Italy, and Portugal report higher rates (over 40%). Conversely, Ireland, Luxembourg, and Sweden boast high tertiary education rates (over 45%), with Romania, Italy, Croatia, and the Czech Republic at the lower end (below 25%). Notably, over 80% of Europeans aged 25–54 years have at least upper secondary education, a contrast to the 68% of the 55–74-year-old age group who have achieved the same level (EUROSTAT 2023).

The 2021–2030 policy framework targets supporting individuals in attaining higher education levels to meet the growing demand for skilled workers, thereby driving innovation and economic growth and improving citizen well-being. The European Education Area Resolution (2021–2030) aims for a minimum of 45% of the 25–34-year-old age group to have tertiary education by 2030 (European Commission 2023).

Investment in human capital should extend beyond primary or secondary education, offering European citizens opportunities for tertiary education and continuous professional training. The Bologna (European Commission 1999) and Copenhagen (European Commission 2002) have facilitated a pan-European higher education area, enhancing mobility, qualification recognition, and cooperation in vocational education (European Commission 1999; European Commission 2002). This fosters exchanges among diverse European cultures and traditions (Powell and Solga 2010).

Education's role in economic growth can be seen in three ways: as a driver of growth, a result of growth, or having a bidirectional relationship with growth. It is pivotal for labor market integration and prosperity, transferring knowledge and ideas from the educated to businesses and projects and fostering innovation (Dudzevičiūtė and Šimelytė 2018). European strategies concur that higher education levels are vital for employment and prosperity, contributing to equitable territories both economically and knowledge-wise (Caleiro 2018).

However, the positive impact of education on territorial growth is not always clearcut. Endogenous growth theory posits that higher skills and education levels positively influence physical capital productivity and growth (Marquez-Ramos and Mourelle 2019). Yet, local conditions such as job scarcity or low wages can weaken the education–growth nexus, prompting skilled labor migration (Quintano et al. 2018; Pinho et al. 2015).

Migration trends among university graduates reveal that the majority remain in their region of study, though those who migrate for education are more likely to relocate postgraduation (Ciriaci 2014). Factors influencing migration include university reputation,

labor market accessibility (Aronica et al. 2023; Dotti et al. 2013), and local economic conditions (Hermannsson et al. 2019). Lagging regions often become net exporters of students and fail to attract those from other areas. This one-way migration of skilled individuals from poorer to richer regions—selective migration—exacerbates spatial inequalities and perpetuates regional disparities in human capital (Cerqua et al. 2022; Guzi et al. 2021; Sardadvar and Vakulenko 2021).

*2.3. The Vicious Circle of Skilled Worker Migration in Southern Italy*

Historically, the migratory outflows from the Italian Mezzogiorno have shaped the socio-economic fabric of southern Italy. The 150-year timeline reveals a transition from the predominantly uneducated migrants of the early 20th century to the present-day exodus of educated young adults (Etzo 2011). This shift, particularly evident since the latter half of the 20th century, has been exacerbated by economic crises and a decrease in labor-intensive industries (Bonifazi et al. 2021). Since the 1990s, a resurgence in emigration has been noted, predominantly among those with secondary and higher education, propelled by disparate economic development between the southern regions and the more affluent central and northern Italy (Etzo 2011; Odoardi and Muratore 2019). Today's migratory flows are characterized by the internal and international relocation of individuals seeking superior educational and professional opportunities, contributing to a self-perpetuating cycle of territorial dualism and the enrichment of foreign labor markets at the expense of local development (Aronica et al. 2023; Basile et al. 2019).

The drivers of this migration are multifaceted, with educational and employment opportunities being primary motivators for those seeking regions where their skills are recognized and rewarded (Ferrara et al. 2018). The statistical narrative provided by the Instituto Nazionale di Statistica (ISTAT) for 2018 (ISTAT 2018) delineates a stark portrait of interregional transfers, signifying a substantial migration of the educated populace from the Mezzogiorno to more prosperous regions, further depleting the south's human capital (Statista 2019).

While labor mobility is traditionally seen as a mechanism for economic equilibrium, the Italian scenario deviates from this model due to the selective nature of the migration, which prioritizes highly qualified individuals. This selective emigration, as opposed to a homogenous distribution of workers, reinforces the dualistic nature of the Italian economy, exacerbating regional inequalities (Basile et al. 2019; Ballatore and Mariani 2019). The intervention of public policies has, paradoxically, not mitigated but seemingly intensified these disparities, perpetuating a negative cycle of talent outflow and contributing to the competitive advantage of the northern regions (Coniglio and Prota 2017).

In a critical analysis of regional policy effectiveness, Coniglio and Prota (2017) highlighted the divergent outcomes in Puglia and Basilicata, regions that have implemented cohesion policies to stem the brain drain. Basilicata's incentives for postgraduate studies have been undercut by a sustained outflow of human capital, while Puglia's bid to synchronize education with local industry demands has seen limited success, with many skilled workers still facing underemployment or job insecurity. Kerr et al. (2016) underscored the potential benefits of return migration, which can significantly contribute to social and human capital by reintegrating global knowledge and resources; however, such potential remains largely unrealized due to persistently low return rates.

On an individual level, experiences abroad offer invaluable exposure and opportunities for collaboration (Avveduto 2012). Nevertheless, these experiences do not inherently counteract the inequalities between peripheral and core regions. The propensity for international migration among the higher-skilled cohort, including those trained with regional and European funding, remains pronounced (Aronica et al. 2023).

The consequences of this ongoing skilled migration are profound, stripping regions not only of their human capital but also of their social and cultural identity. The resulting depletion in territorial capital is accompanied by the diminution of skilled labor, leading to brain drain effects and lessened contributions from those who remain. These changes

culminate in a multi-faceted loss that spans economic, social, cultural, and institutional dimensions, posing complex challenges for the affected territories (Grebeniyk et al. 2021; Amodio 2022).

Policies aimed at curbing this brain drain must consider the nuanced needs of the local skilled workforce and create conditions conducive to the retention and return of human capital. Such measures would not only counteract the current outflow but could also transform the region into a hub that attracts and nurtures talent, thereby fostering sustainable socio-economic growth.

### 2.4. Calabria Region Context

The Calabria region, covering the extreme southern part of Italy, spans 15,080 km². It has approximately 1,947,131 inhabitants (3.2% of Italy's total population), with an average density of about 128 people per km², which is distributed unevenly across the region (ISTAT 2020). Since 2000, there has been a clear downward trend in population growth (an overall rate of −18%). This decline is attributable to a low birth rate (7.4% in 2020) and a high mortality rate (11.2%), resulting in a natural population decrease (−2.9%) and a lack of territorial attractiveness. The region's attraction index for external migration for study or work was only 27.2% in 2015, significantly lower than the national average of 32.6%, and the net migration rate was negative (−14.1%) (AdminStat 2020).

In 2019, Calabria's GDP per capita was 33.61 billion euros, accounting for −1.88% of Italy's total GDP for that year (Statista 2019). The unemployment rate stood at 21.9%, with the rate among young workers reaching a peak of 48.6%. The general wealth of the inhabitants is considerably low (ISTAT 2021), with a poverty rate significantly higher (30.6%) than the national figure (11.8%). The percentage of individuals in relative poverty (34.6%) is more than double that of the entire country (15.0%) (ISTAT 2020). These conditions have positioned Calabria as the third-lowest Italian region in terms of employment rate and the highest in unemployment (Iaquinta et al. 2020). The region is experiencing a worsening phenomenon of "human desertification" (Musolino et al. 2020), driven by a low level of industrialization (in 2017, companies in Calabria represented only 2.5 percent of the national total (ISTAT 2020)), a noncompetitive production system (characterized by low levels of internationalization and innovation), poor infrastructure and transport accessibility, and inadequate development in information and communication technologies (Musolino et al. 2020).

Based on the literature reviewed, it becomes essential to evaluate the effectiveness of the European Funds (ESF) aimed at enhancing the skill set of Calabrian graduates, thereby improving their employability prospects within their native region and encouraging the repatriation of those employed outside the region.

The effectiveness of the ESFs was evaluated through a bottom-up (micro-level) approach using a tailor-made questionnaire aimed directly at assessing whether or not the objective of disbursing ESFs in the Calabria region was achieved.

The following research questions are posed as the basis for the empirical verification process within this study:

- Research Question 1 (RQ1): To what extent does the perceived quality of the knowledge, competencies, and skills acquired through master's programs funded by the ESF, as well as the employment situation at the start of the postgraduate program, positively influence the likelihood of Calabrian graduates finding employment in their region of origin?
- Research Question 2 (RQ2): Does the perceived quality of the knowledge, competencies, and skills acquired in the master's programs funded by the ESF, along with the employment situation at the start of the postgraduate program, positively influence the decision of Calabrian graduates living abroad to return to Calabria?

These questions are integral in evaluating the strategic deployment of the ESF and serve as a critical component for the formulation of policies aimed at regional educational advancement and employment stimulation.

## 3. Materials and Methods

*Survey Design and Implementation*

To investigate the professional trajectories of master's graduates from Calabria who were beneficiaries of European Union Development Funds (EUDF) for their enrollment in postgraduate programs, a specific survey design was developed. This instrument was based on the adaptation of questions from previous studies, specifically the work conducted by Abreu et al. in 2014 (Abreu et al. 2014). This adaptation was carried out to explore the specific dynamics and post-master's professional outcomes in this particular context. The questionnaire, conducted from March 2021 to June 2022, was addressed to all Calabrian graduates who received ESFs from the Calabria region and was distributed through the website Google Form and distributed by email.

The introductory email explained the research objectives and indicated that the University of Córdoba (Spain) was responsible for data collection and management and that participation was anonymous, free, and voluntary. We also indicated that the results would be published in scientific projects.

The survey questions were developed considering the research questions. The questionnaire was designed in Italian and consisted of three parts. The first part referred to the educational qualifications and characteristics of the master's program attended, while the second part consisted of 12 questions about the overall satisfaction with the master's program, the subsequent professional status, and the future perspective regarding a possible return to Calabria. A 5-point Likert scale was used for these questions, which were dichotomized to use endogenous and exogenous variables in the logit models. Finally, the third part referred to the sociodemographic characteristics of the respondents. The administered questionnaire is reported in the Table S1 from Supplementary Materials.

The questionnaire's validity was determined by consensus among the authors and by a pretest with eight randomly selected graduates from Calabria. The length of the survey, the appropriateness of the questions, and the equivocality of the meanings were evaluated.

This study was designed to empirically assess the efficacy of EU Structural Funds allocated from 2014 to 2020 to the Calabria region. It specifically scrutinizes the funds awarded to Calabrian graduates for enrollment in master's programs within and beyond regional boundaries, aiming to enhance their professional skills and competencies. Our analysis was bifurcated into two primary segments. Initially, we delineated the attributes of Calabrian graduates who availed ESFs for higher educational pursuits, such as master's degrees, locally or outside Calabria. This examination encompassed their perceptions and responses regarding various aspects of the master's programs, including the quality of education and the breadth and depth of skill acquisition.

The study employed two logistic regression models to analyze the effects of ESFs on the employability of graduates from Calabria. The first model examined the impact of these funds on the employability of graduates who remain within Calabria. The second model evaluated how EFs influence the inclination of Calabrian graduates who are currently working and living outside the region to consider returning to their place of origin. Statistical analyses were conducted using STATA18 (Stata Corp., College Station, TX, USA).

This bifurcated approach aimed to understand both the direct regional benefits of ESFs on local graduates and their broader influence on the geographic mobility decisions of the Calabrian diaspora. Table 1 presents a detailed description and operationalization of each variable used in our analysis, laying the groundwork for testing the proposed research hypotheses.

**Table 1.** Study Variables.

| Variable | Type | Description |
|---|---|---|
| Acquired skills during the master's program (ASM) | Categorical | Value 1 if acquired skills during the master's program are perceived useful and 0 otherwise |
| Appropriate wage (AW) | Categorical | Value 1 if the respondent received an appropriate wage (salary commensurate with own skills and knowledge) at the time of master enrolment and 0 otherwise |
| Importance of skills acquired (ISA) | Categorical | Value 1 if the respondent perceived as relevant for his/her employability the skills acquired and 0 otherwise |
| Overall perceived quality (OPQ) | Categorical | Value 1 if the quality of the knowledge acquired is perceived high and 0 otherwise |
| Permanent contract (PC) | Categorical | Value 1 if the respondent had a permanent contract at the time of master's program enrolment and 0 otherwise |
| Perceived utility of subsidy (PU) | Categorical | Value 1 if the graduate believes that the ESF subsidy received was useful and 0 otherwise |
| Time to find a job after master's degree (TIME) | Categorical | Value 1 if the graduate found a job within 6 months after the master's program and 0 otherwise. |
| Working status (WS) | Categorical | Value 1 if the respondent worked at the time of master's program enrolment and 0 otherwise |
| Control Variables | | |
| Age | Quantitative | Age of respondents at master's program enrolment |
| Gender | Categorical | Respondent's gender |
| Population | Categorical | Total number of persons living in the respondent's living area (25,000–50,000, 50,000–75,000, or 75,000–100,000) |
| Disciplinary area | Categorical | The field of respondent' specialization (health area, scientific-technological area, or humanistic-social area) |

The selected determinants affecting the likelihood of work in Calabria following the attainment of a master's degree financed by European funds were divided between graduates' employment conditions at master's program enrolment and graduates' perceptions about the master's program topics.

For both determinants, the following equation was applied:

$$y = \frac{e^{(\beta_0 + \beta_1 x_1 + \beta_2 x_2 + \beta_3 x_3 + \beta_4 x_4 + \beta_5 x_5 + \beta_6 x_6 + \beta_7 x_7 + \beta_8 x_8)}}{1 + e^{(\beta_0 + \beta_1 x_1 + \beta_2 x_2 + \beta_3 x_3 + \beta_4 x_4 + \beta_5 x_5 + \beta_6 x_6 + \beta_7 x_7 + \beta_8 x_8)}} \tag{1}$$

The predicted probability was determined by each logistic regression model.

## 4. Results

The survey garnered participation from one hundred and sixty-six graduates originating from Calabria, representing the entire population of Calabrian graduates who received ESFs. The cohort predominantly consisted of male respondents, constituting 54.2%, with an average age across the sample being 32.5 years. A regional analysis revealed that the majority of these graduates, accounting for 41.6%, hailed from the province of Reggio Calabria. Conversely, the provinces of Crotone and Cosenza had the lowest

representation, yielding 7.8% and 4.8% of responses, respectively. Educational backgrounds were varied, with over 60% possessing degrees in the humanities and social sciences. Graduates with degrees in the scientific and technical fields comprised 27.6% of the sample. A minority of 11.7% held degrees within the health sector.

Regarding academic institutions, most respondents, 66.9%, completed their degrees at universities within Calabria. The remaining one-third were split between those who graduated from institutions outside the Calabria region (8.3%) and those who obtained their degrees online (24.8%). Funding patterns showed that over half of the respondents, specifically 54%, received grants for their first master's program, whereas 45.8% were funded for a second master's degree. The duration of these master's programs was predominantly no longer than 12 months for 82.1% of the participants, while a smaller fraction, 17.9%, engaged in extended programs lasting up to 18 months. The structure of the master's programs for all respondents entailed 1500 h of classroom instruction complemented by 300 to 350 h of practical internship or project work. Notably, less than 1% of the survey participants underwent 400 h of internship training. The detailed characteristics of the survey respondents are comprehensively documented in Table 2.

**Table 2.** Sociodemographic variables.

|  | Frequency | Percentage |
| --- | --- | --- |
| No. of responders | 166 | |
| Age, mean (S.D.) | 32.5 (6.3) | |
| **Gender** | | |
| Males | 90 | 54.2% |
| Females | 76 | 45.8% |
| **Province** | | |
| Catanzaro | 58 | 34.9% |
| Cosenza | 8 | 4.8% |
| Crotone | 13 | 7.8% |
| Reggio Calabria | 69 | 41.6% |
| Vibo Valentia | 18 | 10.8% |
| **Disciplinary Area** | | |
| Health area | 20 | 12.0% |
| Scientific-technological area | 46 | 27.7% |
| Humanistic-social area | 100 | 60.2% |
| **Master University Location** | | |
| In Calabria | 112 | 67.5% |
| Online | 41 | 24.7% |
| Outside Calabria | 13 | 7.8% |
| **Master's Degree** | | |
| First | 90 | 54.2% |
| Second | 76 | 45.8% |
| **Master's Duration** | | |
| 12 months | 136 | 81.9% |
| 15 months | 30 | 18.1% |

*Empirical Analysis Results*

The outcomes of the logistic regression analysis were rigorously evaluated using several diagnostic tests. The Omnibus test yielded a highly significant result ($p < 0.001$), indicating that the models as a whole are statistically significant. Furthermore, the Hosmer and Lemeshow tests, along with the chi-square values for respondents, all returned values greater than 0.1, suggesting a good fit of the models. To address potential multicollinearity issues, a stepwise estimation approach based on the likelihood ratio was employed. This

iterative process ensured that the final models included only those variables that were statistically significant.

The binary logistic regression models outlined in Table 2 were employed in this analysis:

- Model 1 Analysis: The first logistic regression model revealed that the presence of European Funds (ESFs) subsidies and the perceived quality of education in master's programs significantly influence the employment location of Calabrian graduate's postgraduation. Specifically, graduates who received ESFs and appraised the subsidies as beneficial demonstrate a markedly higher likelihood of securing employment within Calabria (β = 3.85, SE: 1.55, $p$ = 0.013). Additionally, a positive association was observed between the perceived high quality of master's program education and the probability of working in Calabria postgraduation (β = 5.39, $p$ = 0.016). Conversely, graduates who rated the skills acquired during their master's program as very high were found to be 50% less likely to work in Calabria after completing their studies (β = −4.35, SE: 2.25, $p$ = 0.053);
- Model 2 Analysis: The second logistic regression model assessed the likelihood that graduates who lived outside of Calabria at the time of enrolling in a master's program would return to the region postgraduation. The results indicate that those who were employed during their master's enrollment had an increased probability of returning to Calabria (β = 2.66, SE: 1.53, $p$ = 0.082). In contrast, for graduates perceiving the ESFs subsidy as beneficial, the likelihood of returning to Calabria after attaining their master's degree decreased significantly by 38% (β = −4.41, SE: 1.93, $p$ = 0.022).

In both models, the control variables of age, gender, population of origin, and academic discipline of the respondents were analyzed and were not significant in any of the models analyzed, so they are not included in Table 3.

**Table 3.** Logistic regression.

|  | (M1) Probability of Working in Calabria | (M2) Probability of Returning to Calabria |
| --- | --- | --- |
| Acquired skills during the master's program (ASM) | −4.36 ** (2.250) | 20.14 (4710) |
| Appropriate wage (AW) | 0.78 (0.680) | - |
| Importance of skills acquired (ISA) | 0.48 (2.013) | 19.60 (2269) |
| Permanent contract (PC) | 0.80 (0.705) | - |
| Perceived utility of subsidy (PU) | 3.85 ** (1.550) | −4.41 ** (1.925) |
| Overall perceived quality (OPQ) | 5.39 ** (2.240) | −40.39 (5228) |
| Time to find a job after master's degree (TIME) | −0.58 (0.729) | 0.17 (1.136) |
| Working status (WS) | −0.17 (0.750) | 2.66 * (1.528) |
| Control Variables | Yes | Yes |
| Constant | Yes | Yes |
| McFadden's $R^2$ | 0.246 | 0.543 |
| Cox–Snell $R^2$ | 0.283 | 0.529 |
| Cragg–Uhler (Nagelkerke) | 0.382 | 0.705 |
| LRchi2 ($p$-value) | 25.28 (0.065) | 39.13 (< 0.001) |

Note: * $p < 0.10$; ** $p < 0.05$

## 5. Discussion

The Cohesion Action Plan presents an opportunity for the educational sector to enhance the capabilities of the youth and address unemployment through the implementation of policies supported by the Structural Funds (Council of the European Union 2021). Knowledge, an essential intangible element for a nation's wealth, has recently taken a central role in economic development (Sánchez Trujillo et al. 2020). High levels of education and proficiency in tacit and transversal skills form the cornerstone for creatively contributing to the development of new products, particularly in crafting their symbolic value.

Furthermore, high educational standards and practical experience elevate organizations' and individuals' ability to assimilate external cognitive elements (Schweisfurth and Raasch 2018). The European Union has prioritized investing in youth education as a key driver for economic development (European Council 2015). Given the escalating importance of knowledge, the ambivalent impact of public investment in higher education in economically lagging contexts becomes particularly significant, as evidenced by recent actions in certain Italian regions. Calabria, for example, has invested substantial resources in the advanced training of its human capital, explicitly aiming to boost regional economic development.

Therefore, this study focuses on two key aspects: On the one hand, it examines whether the funding provided for pursuing master's degrees, which facilitate higher professional qualifications, effectively contributes to the integration of young residents in Calabria into the regional labor market. On the other hand, it analyzes whether completing these master's programs encourages Calabrians living outside the region to return to it.

The first model of this study indicated that the existence of subsidies facilitating young people in undertaking a master's degree and the perceived educational quality of these postgraduate programs enhance the likelihood of job retention among young people in the Calabria region after completing their studies. This relationship suggests that access to quality postgraduate education can enrich education and improve employment prospects for young Calabrians, which is in line with the findings of Dudzevičiūtė and Šimelytė (2018). However, an inverse correlation was also observed between the variable "skills acquired during the master's program" and the likelihood of graduates staying to work in the region. Thus, we infer that as young people acquire greater professional competencies, their intention to remain in Calabria decreases, reflecting a phenomenon consistent with the brain drain theory, as discussed by Morano-Foadi (2006) and Dohlman et al. (2019). These results indicate that although the program promotes the integration of young people into the regional labor market, it fails to effectively retain the more qualified individuals, posing significant challenges for talent retention in the region.

The second model aimed to investigate the possibility of return for Calabrians living abroad who benefited from European Funds. The results reflected the paradox of the multifaceted interaction between these European Funds, the training of graduates, and the local labor market. Although graduates working outside the region perceive that they are more likely to return to Calabria after completing a master's degree, the variable of European Funds' subsidies for master's studies has a more intensely negative effect on the inclination to return. This suggests a more complex relationship between fund perception and migration and return decisions, likely influenced by factors such as career prospects and employment opportunities available elsewhere (Hermannsson et al. 2019) that are absent in Calabria.

Participation in master's programs has enabled Calabrian graduates to gain significant experiences, as postgraduate course content is vital for job placement and future employment opportunities. The master's experience proves beneficial in acquiring technical-professional competencies for job hunting. However, these skills and experiences are not aiding in securing employment within Calabria, where a clear disconnect between employability and completed master's degrees is evident, as suggested by Kerr et al. (2016). This contradiction might stem from a lack of specific job opportunities suited to their qualifications.

Beneficiaries of regional policies have found employment outside their birth regions, often in economically more dynamic and mature contexts that better absorb highly qualified human capital (Pinho et al. 2015). This phenomenon, a physiological brain drain, occurs with training investment in economically peripheral regions like Calabria and should be anticipated and managed with other active employment policies in the region, aligning with the thematic objective of the 2014/2020 ESF investment in the policy measure's application region (European Commission (ESF 2014/2020) n.d.).

The reasons for the disconnect between fostering increasingly high competencies and the region's absorption of new qualified labor must be understood as a result of a multidimensional phenomenon. On one hand, the region promotes initiatives to encourage graduates to enhance their knowledge and skills; on the other, it appears ineffective in supporting the labor market's demand for new and qualified workers. This issue is influenced by factors including high tax pressure, crime rates, lack of significant infrastructure, slow bureaucracy, and inefficient social services and support policies (Lombardo and Falcone 2011; Lucatelli and Peta 2010).

We suggest that regional development policymakers aiming to promote regional developments through "human capital shocks" should consider integrating regional intervention programs more fully, namely by fostering synergies between ESF-funded policies and those supported by the European Regional Development Fund (ERDF), which is responsible for promoting business competitiveness even though it also promotes smart specializations of territorial contexts. This integration, which has already been requested in the new EU Structural Funds programming, could address both labor supply and demand sides, considering a human capital approach along with growing Calabrian companies' capacity to absorb human capital and, more broadly, qualifying a local and regional innovative business ecosystem.

This challenging situation in Calabria is further complicated by increased international mobility, reducing the economic appeal of less developed regions in favor of those with stronger economies. Indeed, a vicious cycle exists between promoting a knowledge economy and low educational levels: The lack of qualified human capital diminishes development opportunities, and inversely, in a structurally backward economy, investments in training highly qualified workers are negated by their propensity to migrate (Docquier et al. 2009).

The consequences of this paradoxical situation and the historical problems from which the region of Calabria suffers are enormous. Firstly, the continuing massive loss of young human resources impoverishes the entire social fabric and reduces the chance of coherent and integrated social, financial, and technological development, as is recommended instead in the Cohesion Policy. Secondly, the territories lack new entrepreneurial initiatives, widening the gap with the advanced European regions. Thirdly, the region performs poorly in terms of investment attractiveness, which exacerbates the vicious circle described above. From the results of this work, and assuming a broader view of the issue, it is clear that ESFs disbursed in this way and without proper control of the actual impact on the territory are not only ineffective in reducing inequalities but also create new inequalities as they can worsen the socio-economic situation and thus contribute to widening the gap between European regions. The introduction of such a policy should be planned according to a different approach in which not only the needs of the people but also the characteristics and needs of the territory should be thoroughly assessed so that the policy is beneficial in the shortest possible time not only for the people but also for the territory that provides the resources.

This study has limitations. Primarily, the population mainly comprised participants from the provinces of Catanzaro and Reggio Calabria, with Cosenza, the most populous Calabrian province and home to one of the three Calabrian universities, being underrepresented. Further investigation is needed into the lack of surveyed graduates residing in this province. Secondly, the survey was conducted after the COVID-19 pandemic spread, which prompted many Calabrian citizens to move from northern regions, reducing

emigration rates. Thirdly, the actual reasons driving some respondents not to return to Calabria were not specifically investigated. Further studies should delve into the causes of Calabria's low rate of employed graduates and the negative repercussions due to the loss of more qualified workers. While the employed quantitative methodology offers significant insights, integrating qualitative methods such as interviews and case analysis could provide a deeper understanding of graduates' experiences and perceptions.

Future research should focus on deepening our understanding of the factors influencing the low rates of return among highly qualified graduates to less developed regions such as Calabria. It is also essential to extensively examine the interplay between EU funding, skill development, and labor migration in various developing European areas. Such studies could provide crucial insights for the formulation of more effective policies and strategies in the field of regional development.

## 6. Conclusions

This study conducted a comprehensive analysis of the EU Structural Funds in Calabria, revealing a complex landscape. The results show that the ESFs have been effective in improving the skills and education of graduates in the region, thereby facilitating their integration into the labor market. However, these same funds have generated unintended effects, such as the intensification of brain drain and the increase in regional disparities. Graduates, upon improving their skills and competencies, tend to seek employment opportunities in regions with more dynamic economies, leaving Calabria in an unfavorable cycle of loss of talent and qualified human resources.

Contrary to the expectations raised by Calabria's policymakers through the disbursement of EFs, this dynamic acts as a vicious circle in which the Calabria region itself continues to lose economic and human resources and increases the disparities with the more advanced regions through the same measures it takes to reduce these disparities.

From a future policy perspective, the need for more holistic and locally adapted approaches becomes evident. Employment policies must go beyond skill development and consider creating an attractive working environment that retains graduates in the region. This includes incentives for businesses investing in innovation and technology as well as support for entrepreneurship, especially among highly qualified workers. These measures could help transform the improvement in education and skills into a driver of local growth and development.

Regarding the impacts on economic policy and society, the results underline the importance of regional economic planning that considers both human capital and the needs of the local labor market. Brain drain represents not only a loss of investment in education but also carries a significant social and economic cost for the region. Economic policies must, therefore, focus on closing the gap between the supply of skills and the demand of the labor market, thereby promoting a more balanced and sustainable socioeconomic development.

This study highlights the complexity of the effects of the ESF in regions like Calabria. While the funds have improved education and skills, they have also contributed to significant challenges in terms of talent retention and regional cohesion. Adapting EU cohesion policies to local realities, with an integrated approach that combines skill development and employment opportunities, is crucial to turn investments in human capital into real assets for regional development.

**Supplementary Materials:** The following supporting information can be downloaded at: https://www.mdpi.com/article/10.3390/economies12010010/s1. Table S1: Questionnaire variables.

**Author Contributions:** Conceptualization, A.B. and G.A.M.-F.; methodology, A.B., formal analysis, M.L.G., data curation, M.L.G.; writing—original draft preparation, A.B., M.L.G., and G.A.M.-F.; writing—review and editing, J.E.R.-R. and G.A.M.-F.; visualization, J.E.R.-R. and G.A.M.-F.; supervision, G.A.M.-F. All authors have read and agreed to the published version of the manuscript.

**Funding:** This research received no external funding.

**Institutional Review Board Statement:** Not applicable.

**Informed Consent Statement:** Informed consent was obtained from all subjects involved in the study.

**Data Availability Statement:** Data supporting this study are included within the article.

**Conflicts of Interest:** The authors declare no conflicts of interest.

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
