# Peer review of "The Dynamics of Fund Absorption: Evaluating the Efficacy of EU Structural Funds in Mitigating Regional Inequalities—Calabrian Case"

_economies, doi:10.3390/economies12010010_

Round 1

Reviewer 1 Report

Comments and Suggestions for Authors

The article presents a very interesting perspective on the issue of  regional policy internal coordination. It can be used as an argument for the, increasingly promoted, shift away from sectoral (silo) policies towards a holistic consideration of the social costs and benefits of public investment.   The only minor reservation is the lack of broader justification of the variables adopted. At the same time, the author(s) reasonably point out the necessary additions and shortcomings of the study, which are necessary for a complete analysis.  Nevertheless, the article is an interesting contribution to the discussion on the effectiveness of European funds.

Author Response

Dear Reviewer,

I deeply appreciate your valuable comments and suggestions regarding our article. It is gratifying to know that the study has been recognized as an interesting contribution to the debate on the effectiveness of the European Structural Funds, particularly in terms of regional policy and its internal coordination.

Regarding your observation on the lack of a broader justification for the adopted variables, I acknowledge the importance of this perspective. The selection of variables was based on specific criteria related to the work of Abreu et al. (2014) (lines 301-305), as well as on the relevance and availability of data, also considering the inherent limitations of the study's scope. However, I agree that a more detailed justification could further strengthen the analysis. In future work, I intend to delve deeper into this aspect, possibly expanding the theoretical framework to include a more thorough discussion on the choice of variables.

As for future perspectives and relevance for economic policy and society, this study aims to pave the way for more detailed investigations that examine the multifaceted effects of EU funds in different regional contexts. The goal is to provide a framework for policymakers to design more effective interventions that not only focus on human capital investment but also consider labor market dynamics and regional economic development.

Finally, it is important to highlight that this study aligns with the growing trend of moving away from isolated sectoral policies towards a more holistic approach that considers the social costs and benefits of public investment. Through this approach, we aspire to contribute to a more nuanced understanding of how EU funds can be used more effectively to foster regional cohesion and development.

Once again, I thank you for your comments and hope that the proposed modifications and clarifications are satisfactory.

Sincerely,

Reviewer 2 Report

Comments and Suggestions for Authors

The article is of interest to regional development with direct reference to the impact that European funds have on future development.

There are some issues that should be taken into consideration by the authors to improve its quality:

1. The assumptions should be introduced

2. The Methodology section will be called Materials and Methods

3. In this section, information about the software used to analyze the responses, information about the selection of respondents, and whether there is representativeness for the sample studied should be integrated. Please explain if the questionnaire was sent by email (see line 279).

4. The Discussion section should include information on how results can be interpreted in perspective of previous studies and of the working hypotheses.

5. Conclusions should be more extensive and summarize the paper's findings, achievement of objectives, and future research.

Good luck!

Author Response

Dear Reviewer,

Thank you for your constructive feedback on our article. We have carefully considered each point and made the necessary revisions to enhance the quality of our work. Below are our responses to your suggestions:

  1. Introduction of Assumptions
    Authors’ reply: In the introduction, we have now incorporated the assumptions and expected results of this study, focusing on the application of European Funds (ESFs) to the upskilling of Calabrian graduates.
  2. Methodology Section Title
    Authors’ reply: The Methodology section has been renamed to "Materials and Methods" to better reflect the content and approach of our research.
  3. Details on Methodology
    Authors’ reply: We have added detailed information about the software used (STATA18) in the analysis (see line 341). The questionnaire, administered via Google Form, was distributed by email. In terms of representativeness, we contacted the entire population of Calabrian graduates and included all those who received ESFs in our study (see lines 307-308).
  4. Discussion Section Content
    Authors’ reply: The discussion section now includes information relating our results to those of previous studies, providing a comparative perspective and alignment with working hypotheses (see lines 429-471).
  5. Conclusions Section
    Authors’ reply: The conclusions section has been expanded to more comprehensively summarize the findings of the work and the achievement of the objectives (see lines 533-560). We have also included a discussion on future lines of research in the Discussion section (see lines 524-529).

We hope these revisions address your concerns and improve the overall quality and clarity of our manuscript. We appreciate the opportunity to refine our work based on your valuable insights.

Sincerely,

Reviewer 3 Report

Comments and Suggestions for Authors

Dear Authors,

Before the possible publication, a few review questions should be answered. Please find the questions:

  1. What is the primary focus of the study, and which region is under consideration for assessing the impact of the EU Structural Funds (ESFs)?

  2. Why have ESFs been directed towards supporting Calabrian graduates, and what is the intended outcome of this support?

  3. How does the study measure the impact of ESF subsidies on graduates in Calabria, and what methods are employed to explore the effectiveness of the funds in achieving their goals?

  4. Summarize the dual effect observed in the results regarding the influence of education quality, EU funding, and advanced skills on graduates' decisions to work in Calabria.

  5. What paradoxical finding does the study reveal about the impact of ESF support on the likelihood of graduates returning to Calabria after working elsewhere?

  6. How does the dynamic described in the findings contribute to a potentially vicious cycle, and what are the implications for regional development in Calabria?

  7. Discuss the broader consequences highlighted in the text, including the potential impact on the socio-economic divide within Europe.

  8. What recommendations, if any, does the study propose to address the unintended consequences of ESFs in Calabria?

These questions aim to assess the reader's understanding of the study's objectives, methods, results, and implications. Please provide the necessary revisions in the draft text according to the indicated questions.

Author Response

Dear Reviewer,

Thank you for your insightful questions regarding our manuscript. Below, please find our responses to each query:

  1. Primary Focus and Region of Study
    Authors’ reply: The objective of the study and the specific region where the impact of the Structural Funds is assessed have been more clearly specified in the Introduction (see lines: 68-71).
  2. Rationale for ESF Support to Calabrian Graduates
    Authors’ reply: The reasoning behind directing ESF support towards Calabrian graduates and the intended outcome of this support have been detailed in the Introduction (see lines 58-65 for the rationale and 78-82 for the expected outcome).
  3. Methodology for Measuring Impact
    Authors’ reply: We have expanded the Methodology section to include more information on how the impact of ESF subsidies on graduates in Calabria is measured and the methods employed to explore the effectiveness of the funds (see lines: 282-284).
  4. Dual Effect Observed in Results
  5. Paradoxical Finding on Graduates Returning to Calabria
  6. Implications for Regional Development in Calabria
    Authors’ reply (for questions 4, 5, 6): In the Conclusions, we have elaborated on the dual effect observed in the results, especially focusing on the influence of improved education and training on graduates’ decisions. We also discuss the paradoxical finding regarding the impact of ESF on graduates’ likelihood of returning to Calabria and its implications for regional development (see lines 533-544).
  7. Broader Consequences and Socio-economic Divide in Europe
    Authors’ reply: The broader consequences of our findings, including the potential impact on the socio-economic divide within Europe, have been discussed in the Discussion section (see lines 496-511).
  8. Recommendations to Address Unintended Consequences
    Authors’ reply: We have included our recommendations in the Conclusions to address the unintended consequences of ESFs in Calabria (see lines 545-551).

We hope these responses adequately address your queries and further clarify our study's contributions. We are grateful for the opportunity to enhance our manuscript through your feedback.

Sincerely,

Reviewer 4 Report

Comments and Suggestions for Authors

Dear Authors,

Thanks for contributing to the efficacy of EU Structural Funds and mitigating regional inequalities discourses. The work has addressed a very significant subject matter with a clear literature base and analytical methods. However, there are some aspects that are still not very clear and make the paper hard to comprehend. I have detailed these comments for your keen consideration and revision for improvement.

1. Please include a statement with a research gap or problem statement in the abstract, along with a recommendation or a concluding statement to make the abstract complete.

2. Regarding lines 103-105, From a neoclassical perspective, Structural Funds are vital for budgetary allocations 103and distribution within disadvantaged EU areas. and “Conversely, endogenous growth theories emphasize the long-term role of public policies in growth through human capital, innovation, and knowledge investments.”

I'm confused, these sentences don't seem to be in opposition, and there are logic issues.

3. Please further clarify the paper's innovative points and marginal contributions.

4. Please arrange the paper's structure more effectively. I believe the introduction section is too lengthy, and I suggest the authors divide it into three parts: introduction, literature review, and theoretical analysis.

5. Why choose Calabrian as the study subject? The authors should provide further explanation within the paper.

6. I suggest that the authors should integrate theoretical analysis into the formulation of research questions, rather than abruptly presenting these two research questions in the "Objective of the Study" section of this paper.

7. Authors are suggested to improve the conclusion section as well since it broadly handled and should be very concrete for the description of the results followed by the policy. How your study can be benefited for society? I would suggest a separate policy section.

8.  Add the limitations and future recommendations of the study.

Comments on the Quality of English Language

The authors can further review the context to ensure that the sentences are more coherent. This will help improve the readability and logical consistency of the paper.

Author Response

Dear Reviewer,

Thank you for your thorough review and valuable feedback on our manuscript. We have taken your comments into consideration and made appropriate revisions to enhance the clarity and quality of our paper. Below are our responses to each of your points:

  1. Abstract Revision
    Authors’ reply: A problem statement has now been included in the abstract, and the wording of the final statement has been improved for better clarity and completeness.
  2. Clarification on Theoretical Perspectives
    Authors’ reply: We have revised lines 103-105 to correct the logical flow. The term “Conversely” has been changed to “Similarly,” as this was an error in presenting the two theories. We apologize for the confusion caused (see line 127).
  3. Paper's Innovative Points and Contributions
    Authors’ reply: The innovative points and marginal contributions of the paper have been further clarified in the Introduction section (lines 72-77).
  4. Paper Structure and Introduction Section
    Authors’ reply: The introduction has been restructured into three distinct parts: Introduction, Literature Review, and Theoretical Analysis, to enhance readability and coherence.
  5. Rationale for Choosing Calabria as the Study Subject
    Authors’ reply: An explanation for selecting Calabria as the study subject has been added to the paper (see lines 58-64).
  6. Integration of Theoretical Analysis with Research Questions
    Authors’ reply: We have integrated the theoretical analysis into the formulation of the research questions. This integration has been made more explicit in the paper (see line 278).
  7. Improvement of the Conclusion Section
    Authors’ reply: The Conclusion section has been revised to be more concrete and specific, including more details about the dual effect observed in the results and the recommendations derived from the study (lines 533-558).
  8. Limitations and Future Recommendations
    Authors’ reply: We have added future recommendations in the Discussion section (lines 524-529) and addressed the study's limitations (see lines 512-513).

Comments on the Quality of the English Language
Authors’ reply: We have further reviewed and refined the language and sentence structure to enhance coherence and clarity throughout the paper.

We appreciate your input, which has been instrumental in improving our manuscript. We hope that these revisions adequately address your concerns and contribute to the overall quality of the work.

Sincerely,

Reviewer 5 Report

Comments and Suggestions for Authors

The manuscript entitled: "The dynamics of fund absorption: evaluating the efficacy of EU Structural Funds in mitigating regional inequalities. Calabrian " needs some improvements, which we present below.

1. Abstract: The authors need to revisit presenting the methodology used and the conclusions reached by the empirical study.

2. Introduction: Authors must present the objectives of this investigation. This means that items 1.1. up to 1.5 should appear in an independent point called a literature review.

3. In the literature review, if the authors agree with our comments, they must include more current references.

4. Methodology: It was interesting to present and justify the empirical hypotheses according to the literature.

5. Results: The authors could present some descriptive statistics and correlations between the variables used in the investigation, following Table 2.

6. Conclusions: These need to be extended by referring to the results found, future perspectives, and impacts on economic policy and society.

Author Response

Dear Reviewer

Thank you for your thoughtful comments and suggestions on our manuscript entitled: "The dynamics of fund absorption: evaluating the efficacy of EU Structural Funds in mitigating regional inequalities in Calabria." We have made the following revisions in accordance with your feedback:

  1. Abstract Revision
    Authors’ response: We have revised the abstract to better present the methodology used and the conclusions reached by the empirical study.
  2. Introduction and Literature Review
    Authors’ response: The introduction is now clearly divided into three parts: Introduction, Literature Review, and Theoretical Analysis, aligning with your suggestions.
  3. Updated References in Literature Review
    Authors’ response: We have updated some of the references in the literature review to include more current studies and perspectives.
  4. Methodology and Empirical Hypotheses
    Authors’ response: Although our research focused on research questions rather than hypotheses, we have clarified in the Theoretical Analysis section how these questions arise from the literature review (see line 278). Additionally, in the Survey Design and Implementation section, we have detailed the previous research that informed our survey (lines 301-305).
  5. Presentation of Descriptive Statistics and Correlations
    Authors’ response: We appreciate this suggestion. However, given the nature of our study and methodological approach, introducing correlation coefficients between variables was deemed unnecessary. As the variables in the study are dichotomous (shown in Table 1), traditional correlation analysis may be inappropriate. We have focused on the study's primary aim in our presentation, assessing specific elements like work status and perceived utility of the subsidy and their impact on the probability of working or returning to Calabria.
  6. Extension of Conclusions
    Authors’ response: In the Conclusions, we have expanded our discussion to include more information about the dual effect observed in the results, particularly focusing on the influence of improved education and training. We have also included recommendations based on the results of the study (lines 533-558).

We hope these revisions address your concerns and enhance the clarity and depth of our manuscript. We are grateful for the opportunity to improve our work through your valuable insights.

Sincerely,

Round 2

Reviewer 2 Report

Comments and Suggestions for Authors

I would suggest to the authors to move the content of section 3, as a separate paragraph at the end of section 2. I believe that the title '3. Theoretical analysis' is not necessary. 

Reviewer 4 Report

Comments and Suggestions for Authors

I am satisfied with the author's response, so the manuscript can be accepted.

Reviewer 5 Report

Comments and Suggestions for Authors

The authors tried to review the manuscript and sought to respond to the reviewers. So the manuscript can be accepted.